# An algorithmic approach to detect generalization in sketch maps from sketch map alignment

Charu Manivannan *, Jakub Krukar, Angela Schwering

Institute for Geoinformatics, University of Münster, Münster, Germany

* c_mani01@uni-muenster.de

## Abstract

Sketch maps are valuable tools used across various disciplines including spatial cognition, environmental psychology, and spatial reasoning. A common approach to evaluate sketch maps in research is to align and compare them with metric maps. However, sketch maps are highly abstract and contain generalized information causing difficulty in their alignment. Current approaches to study sketch maps cannot handle generalized information. They require a one-on-one correspondence between features in the metric map and features in the sketch map. But memory is often generalized. This paper makes two contributions to the research on sketch maps: (i) we present an algorithmic approach to detect generalization in sketch maps (ii) we present an online tool that creates a generalized metric map corresponding to features in sketch maps. Previously, we identified nine types of generalization in sketch maps. In this paper, we develop formal operators to detect these generalizations and implement them as an online tool. We evaluated our algorithm with a set of 11 sketch maps containing 84 instances of generalization. The results indicated that our algorithm consistently detects instances of generalization in sketch maps.

## 1 Introduction

Sketch maps are hand-drawn abstract representations of spatial information considered to be externalization of one's acquired spatial knowledge. They have wide-ranging applications across multiple domains. In environmental psychology, sketch maps are used to analyze how individuals perceive their environment [1–3]. Oftentimes, they are integrated with traditional GIS approaches to do spatial analysis such as analyzing human emotions [4–6]. They are also used as a Volunteered Geographic Information tool to collect data for urban planning, disaster management, etc. [7–10]. One common process that sketch maps go through is manual alignment with metric maps or any target environment. This is because sketch maps, being hand-drawn and subject to individual differences, are often incomplete, distorted, fragmented, and generalized representations of the environment. Therefore, a 1:1 alignment with a metric map is crucial to ensure that sketched data are understood and comparable in a standardized manner. Manually aligning corresponding features is a tedious and error-prone process because in real-world experiments it is rare that all features in a sketch map have a 1:1 relationship with features in a

**Data Availability Statement:** To ensure reproducibility, the data used for the analysis and the results from the online tool are deposited in the corresponding author's OSF project (https://doi.org/10.17605/OSF.IO/X8KP7) linked to the github

repository – "Generalization in sketch maps" (https://github.com/CharuManivannan/Generalization-in-Sketch-map). The code used to implement the algorithm can be found at https://github.com/ifgi-sil/SketchMapia-SoftwareSuite.

**Funding:** This work was supported by the German Research Foundation (SCHW1372/7-3, "Sketchmapia") and the Swiss National Science Foundation (Sinergia 202284, "3D Sketch Maps"). We also acknowledge support from the open access publication fund of the University of Münster. The funders had no role in study design, data collection and analysis, decision to publish, or preparation of the manuscript.

**Competing interests:** The authors have declared that no competing interests exist.

metric map. During sketching, people do not follow any consistent rules of generalization, resulting in an inconsistently generalized sketch map wherein some features are not generalized, some are omitted, some features are merged, and others are collapsed to a point. An unbiased alignment requires systematic identification and resolution of these generalizations in sketch map.

Our earlier work [11] on generalization in sketch maps identified nine distinct types of generalization [11]. In the current paper, we present an algorithmic approach to automatically detect and resolve these generalizations which may contain 1: 1, 1: M, or even M: N alignments. To resolve the 1: M or M: N alignments, we generalize the metric map to match the specific sketch map resulting in a 1: 1 alignment. Our approach to deal with generalization can be used in many different existing sketch map analysis approaches. We tested our algorithm on a set of 11 sketch maps containing 84 generalized features and found that our algorithm detected all generalizations, unlike human raters who failed to identify generalizations in complex sketch maps. The algorithm has been implemented as an online tool and integrated with the sketch map analysis method "SketchMapia" [12].

We present sketch map analysis as a use case where generalization in sketch maps is a significant problem and requires consistent identification. Following this, we provide an overview of related works (section 2). A detailed explanation of our algorithmic implementation can be found in section 3. Evaluation of our algorithm is presented in section 4 followed by the conclusion (section 5).

## 1.1 Use case 1: Sketch map analysis

Several studies have demonstrated that sketch maps are reliable predictors of spatial learning and navigation performance [13–15]. While sketch map contains extensive information on an individual's spatial knowledge, analyzing this information is complicated. Some of the existing methods for sketch map analysis are measuring completeness [1] by manually counting features, qualitative accuracy by checking whether the same topological relations hold between aligned features in sketch map and metric map [16, 17], and quantitative measures such as overall configurational accuracy [18, 19] using bi-dimensional regression. All these methods are based on a pairwise comparison requiring a one-on-one alignment between features in the sketch map and features in the metric map. For example, a landmark drawn as a single polygon with the label "school" in the sketch map must be aligned with a single polygon in the metric map, even if the actual school consists of many buildings and is represented in the metric map as a set of polygons (Fig 1).

The lack of a standardized procedure to resolve generalization in sketch maps may result in biased analysis. Fig 1 shows a metric map along with two probable sketch maps. While some features in these maps have 1: 1 correspondence, other features (highlighted in yellow) are obviously generalized. It is unclear how these features are to be aligned. For instance, one could align feature "S15" in sketch B labeled as 'shops' with either one of "33", "34", or "35" in the metric map or one could also align "S15" to "33", "34", and "35" together, but then it becomes a 1: M alignment. As there exists no approach to resolve generalization in sketch maps, researchers may focus only on a subset of features that has a one-on-one match ignoring the generalized content leading to information loss. Our algorithm reduces the laborious manual task of resolving generalization by converting group alignments to 1:1 alignment.

## 2 Related works

### 2.1 Generalized features: An issue in sketch map alignment

Generalization is an inherent characteristic of sketch maps that undermines their usability. In the field of sketch map analysis, the simplest method of analyzing a sketch map is to count the

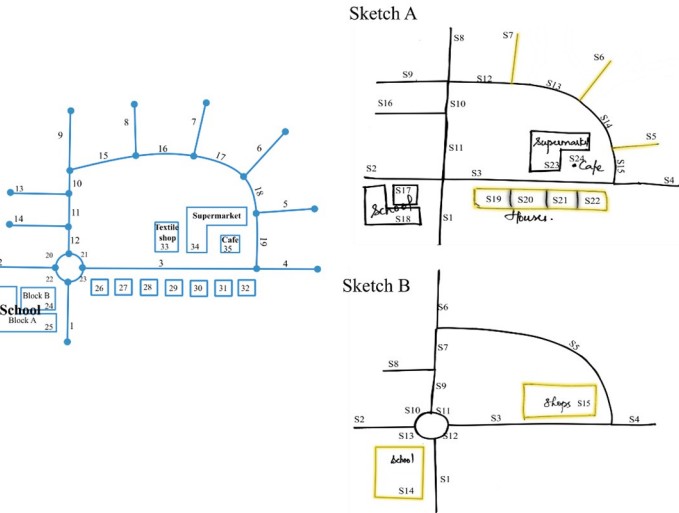

**Fig 1. Two different sketch maps—Sketch A and Sketch B along with their metric map.** Yellow features are generalized leading to an unclear alignment.

number of features drawn relative to the metric map termed "completeness" [1]. Completeness is relatively easy to measure manually without software support. However, when a sketch map contains generalized features, even a simple task such as counting features becomes complicated to do manually in a consistent manner. For instance, in Fig 1, it is unclear whether to consider houses in sketch A as complete or incomplete. Gardony map drawing analyzer [18] is perhaps one of the most widely used software for analyzing sketch maps using bi-dimensional regression [19–21]. While it can handle incomplete sketch maps at present, it lacks the functionality to work with generalized sketch maps. SketchMapia [12] proposes seven qualitative aspects [22] of sketch maps that are not affected by cognitive distortions for measuring the qualitative accuracy of a sketch map. At its current state, SketchMapia also computes these measures only for features aligned in a 1:1 manner. The proposed approach to resolve generalization can be integrated with SketchMapia for converting group alignments to 1:1. To conclude, apart from the fact that only very few software exist for analyzing complex sketch maps, none of the existing software can handle generalized information.

While matching generalized spatial data to detailed spatial data has been done for geo-referenced data [23] e.g. maps of different scales, the same procedure cannot be followed in the case of sketch map, because geo-referenced data have the advantage of similar shapes and other geometric properties for matching. These properties are not preserved in the sketch map. Matching sketched data to georeferenced data relies mostly on qualitative properties [24] between the two maps such as the topology of streets [25], and left-right relations of landmarks to streets [26]. Numerous applications that require aligning spatial data in the sketch map to geo-referenced data have pointed out the issue of limited matching, primarily due to the generalized content within the sketch map [12, 24]. Wolter et. al [27] have explicitly mentioned that M: N alignments in sketch maps need to be addressed for querying spatial databases using hand drawn sketches efficiently. Wallgrün et.al [28] limited their focus on 1:1 alignments for qualitative matching of spatial information due to the computational complexity posed by 1: M or M: N alignments.

Other potential applications of sketch maps are also constrained by the absence of tools to manage generalized content in sketch maps. Geographic Information Systems (GIS) have been

used for storing local indigenous knowledge for flood risk mapping [7, 9, 29], or disaster management tasks [30]. However, this knowledge often exists in fragmented and generalized forms, posing challenges for conventional GIS system designed to work with geo-referenced data. Participatory GIS initiatives [6, 8, 31] frequently employ mapping tools to gather sketch map as overlays on base maps. Integrating data from different sketch maps becomes challenging due to each sketch map being generalized differently. Thus, establishing a standard approach to resolve generalization discrepancies in sketch maps would greatly benefit a wide range of applications relying on sketch maps.

## 2.2 Types of generalization in sketch map

We define generalization as the level of geometric detail in the sketched features. The fewer the details in a sketch map, the more it is generalized. In our previous work [11, 16], we identified nine types of generalization by analyzing 108 sketch maps of a small urban area of approximately 3 sq. km. Out of the nine types, seven types are relevant for resolving generalization in sketch map alignment. The other two are: omission of streets and omission of buildings referring to features present in metric map but omitted in sketch map resulting in 1: 0 alignment which does not necessitate any modification in feature geometry. Fig 2 shows a symbolic representation of the seven identified generalizations with a brief explanation of each generalization type.

As a part of our previous work [11], we conducted a user study to test if individuals can manually detect generalizations in sketch maps if they are made aware of the different generalization types. Ten sketch maps of three different real-world locations were created where generalizations were systematically introduced in the sketch maps. Our 5 participants (4 males and 1 female) showed a high agreement on which features are matching, generalized or missing with an inter-rater agreement scores above 0.90 with a significant p-value less than 0.005. Despite the high agreement rate, there were many instances when participants failed to identify the generalization in the sketch map. The algorithmic approach presented here automatically identifies each of the generalizations and thus outperforms the human raters as sketch maps get too complex.

## 2.3 Existing algorithms on generalization operators

Though generalization in sketch maps is an unaddressed problem, it is extensively researched in the field of cartography. In our earlier work [11], we conducted a detailed study on the differences and similarities between cartographic generalization and sketch map generalization. In McMaster and Shea's work [32] on cartographic generalization, they delineate cartographic generalization into three main tasks: *why* to generalize (the purpose of the map), devising conditions indicating *when* to generalize and identifying the necessary spatial transformations indicating *how* to generalize. In the context of sketch map generalization, the metric map is generalized to have 1:1 alignment (why). The metric map is generalized if a generalization is detected in a sketch map (when). The nine generalization types identified earlier [11] form the generalization operators (how). This manuscript translates our previous theoretically identified operators (how) into an algorithm.

Cecconi [33] draws a difference between generalization operators and generalization algorithms. Generalization operators outline the desired transformation whereas the generalization algorithm provides the step-by-step procedure and method to execute the desired generalization. This also implies that generalization operators are theoretical concepts independent of any particular data model, unlike generalization algorithms that are designed to work with specific input data format. We opted for a vector-based representation of data for our algorithm

| | Generalization Operators | Representation in Metric Map | Representation in Sketch Map | Description |
|---|---|---|---|---|
| (a) | Omission-Merge | | | S1 and S2 in Metric Map getting merged to S' in sketch map due to omission of a street. |
| (b) | Abstraction to show existence in streets | | | A group of streets G in metric map represented as a group of streets G' in sketch map without the possibility to have a one-one match between individual streets. |
| (c) | Junction-Merge | | | Junction J1 and J2 in the metric map merged to form junction J' in the sketch map. |
| (d) | Roundabout Collapse | | | A roundabout R in metric map collapsed to a normal junction R' in sketch map. |
| (e) | Collapse | | | A Building B of the form polygon in the metric map is generalized into a point like building B' in sketch map. |
| (f) | Amalgamation | | | The multi-part building represented by three polygons B1, B2, and B3 in the metric map are merged to form a single building B' in the sketch map. |
| (g) | Abstraction to show existence in buildings | | | A group of buildings G in metric map represented as a group of buildings G' in sketch map without the possibility of having a one-on-one match between individual buildings. |

**Fig 2. Sketch map generalization operators.**

as it is best suited to represent the sketch map data that predominantly consists of linear and aerial objects [34].

A single generalization operator such as collapse or amalgamation can have multiple algorithms [35–38] in cartographic generalization depending on the spatial context, each having its own advantages and disadvantages. In our current work, we demonstrate a single algorithm for each of the seven generalization operators (Fig 2). Some of our generalization operators such as collapse and amalgamation (Fig 2) share names with their cartographic counterparts, but the algorithm is different in the case of resolving sketch map generalization. This is because the intention behind the two types of generalization is significantly different. While algorithms catering to cartographic generalization have to take into account a number of conflicts that might occur during scale reduction, such as proximity conflict, size conflict, etc, our algorithm gives more importance to arriving at 1:1 alignment between the sketch map and the metric map.

## 3 An algorithm to detect generalization in sketch maps

Our algorithm detects generalization in sketch maps from alignment data and applies the necessary generalization operator to the metric map resulting in a 1:1 correspondence between metric map features and sketch map features.

### 3.1 Input data for the algorithm

Features in the metric map and sketch map are fed as input to the algorithm in a vector format (geoJSON): lines for linear features such as streets, polygons for aerial features like buildings, and points for representing one-dimensional features. Every feature in both maps must have a unique ID. The lines representing the street network must be topologically correct, starting and ending at junctions with precise snapping. For a roundabout to be detected, the line segments forming the roundabout loop should have some curvature to it.

Additionally, the algorithm requires alignment data as input. The alignment data is a table (Table 1) containing pairwise IDs of corresponding features in sketch and metric map. Fig 3 shows sketch maps A and B aligned to their metric map and Table 1 the corresponding alignments.

### 3.2 Heuristics of the algorithm

The algorithm derives the cardinal relationship between aligned features (one-to-one, one-to-many, or many-to-many) and the feature types of the aligned features (point, linear, or polygonal features) from the input alignment data and deduces the generalization type following the decision tree in Fig 4. The generalization type is stored in the last column of Table 1.

**Table 1. Input alignment data along with Cardinality and Feature type derived from the input for Sketch A (left) and Sketch B (right).**

| Sketch Map A | | | | | Sketch Map B | | | | |
|---|---|---|---|---|---|---|---|---|---|
| Input alignment data | | Derived data from input | | | Input alignment data | | Derived data from input | | |
| Feature Id (MM) | Feature Id (SM) | Feature type | Cardinality | Generalization Type | Feature Id (MM) | Feature Id (SM) | Feature type | Cardinality | Generalization Type |
| 1 | S1 | Line–Line | 1: 1 | | 1 | S1 | Line–Line | 1: 1 | |
| 2 | S2 | Line–Line | 1: 1 | | 2 | S2 | Line–Line | 1: 1 | |
| 3 | S3 | Line–Line | 1: 1 | | 3 | S3 | Line–Line | 1: 1 | |
| 4 | S4 | Line–Line | 1: 1 | | 4 | S4 | Line–Line | 1: 1 | |
| 5–8 | S5—S7 | Line–Line | M: N | | 15–19 | S5 | Line–Line | M: 1 | |
| 9 | S8 | Line–Line | 1: 1 | | 9 | S6 | Line–Line | 1: 1 | |
| 13 | S9 | Line–Line | 1: 1 | | 10–12 | S7, S9 | Line–Line | M: N | |
| 11 | S10 | Line–Line | 1: 1 | | 13,14 | S8 | Line–Line | M: 1 | |
| 12 | S11 | Line–Line | 1: 1 | | 20 | S10 | Line–Line | 1: 1 | |
| 15–19 | S12-S15 | Line–Line | M: N | | 21 | S11 | Line–Line | 1: 1 | |
| 14 | S16 | Line–Line | 1: 1 | | 23 | S12 | Line–Line | 1: 1 | |
| 24 | S17 | Polygon–Polygon | 1: 1 | | 22 | S13 | Line–Line | 1: 1 | |
| 25 | S18 | Polygon–Polygon | 1: 1 | | 24,25 | S14 | Polygon-Polygon | M: 1 | |
| 26–32 | S19-S22 | Polygon–Polygon | M: N | | 33,34,35 | S15 | Polygon-Polygon | M: 1 | |
| 33 | S23 | Polygon–Polygon | 1: 1 | | | | | | |
| 35 | S24 | Polygon–Point | 1: 1 | | | | | | |

(a) Alignment between Metric map and Sketch map A

(b) Alignment between Metric map and Sketch map B

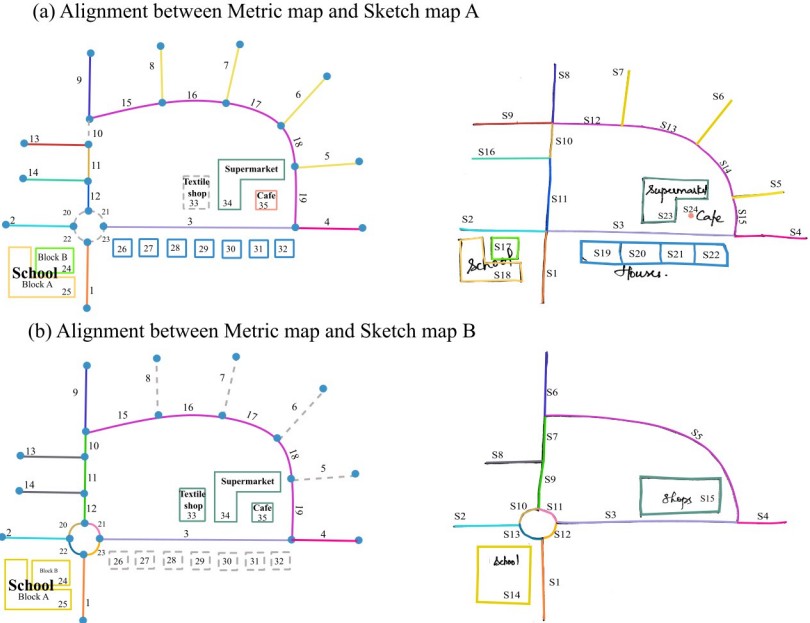

**Fig 3. Visualization of alignment between metric map (left) and sketch maps (right): Features aligned to each other share the same color in the sketch map and the metric map.**

The algorithm requires the aligned features to be of the same feature type. When the aligned features are of different types, the algorithm rejects the alignment and proceeds with the next pair of aligned features. One exception to this is when a polygon feature in metric map is aligned to a point feature in sketch map (refer section 3.2.2 for details).

In the next step, the metric map features are categorized into non-generalized (1:1 relationship), missing (1:0 relationship), or generalized (m:1 or n:m relationship) initially. This is not the final classification as some non-generalized features despite having 1: 1 relationship might have geometrically changed in a way that affects the topology. Such generalizations could be identified only at later stages of the algorithm. The following sections show how the exact generalization type is determined under two cases: Generalization in streets (section 3.2.1) and Generalization in buildings (section 3.2.2), with subcases for each: many-one, many-many, and special cases.

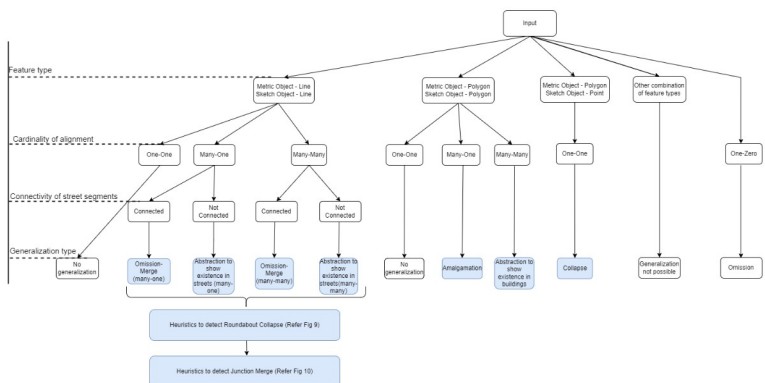

**Fig 4. Decision tree for identifying generalization types from sketch map alignment.**

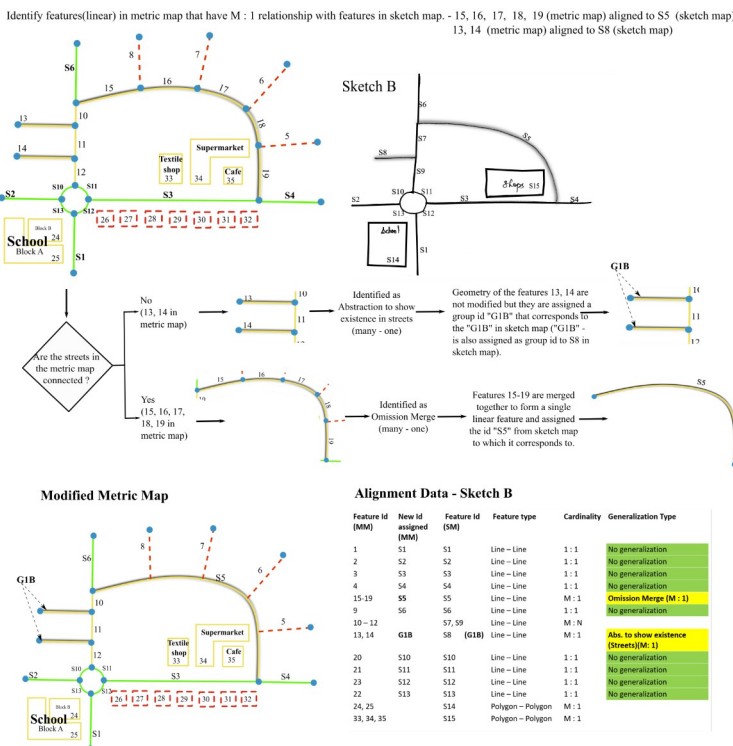

**Fig 5. Algorithm to resolve Omission-merge (many—one) and Abstraction to show existence in streets (many—one).**

**3.2.1 Generalization in streets.** Other than omission, our previous study identified two generalization types related to streets: Omission-merge, Abstraction to show existence, and two other related to streets forming junctions: Junction-merge and Roundabout-collapse. Our algorithm further sub-categorizes omission merge into two: Omission-merge (many-one) and Omission-merge (many-many) and Abstraction to show existence in streets into Abstraction to show existence in streets (many-one) and Abstraction to show existence in streets (many-many).

- **Sub-case: Many-to-one.** The algorithm checks the connectivity of the aligned street segments. If one street segment in the sketch map is aligned to more than one but **connected** street segments in the metric map, then the algorithm detects it as *Omission-merge (many-one)* and merges the connected street segments in metric map into one street segment (Fig 5).

  If one street segment in the sketch map is aligned to more than one but **disconnected** street segments in the metric map, then the algorithm detects it as *Abstraction to show existence in streets (many-one).* The geometry of the street segments is not merged and it is rather treated as one feature in the sketch map corresponding to a group of features in the metric map for further analysis (Fig 5).

- **Subcase: Many-to-many.** On analyzing the connectivity of street segments, if more than one **connected** street segment in the sketch map is aligned to more than one **connected** street segment in the metric map, the algorithm detects it as *Omission-merge (many-many)* and merges the connected street segments to form a single street in both metric map and sketch map resulting in a one-to-one alignment (Fig 6).

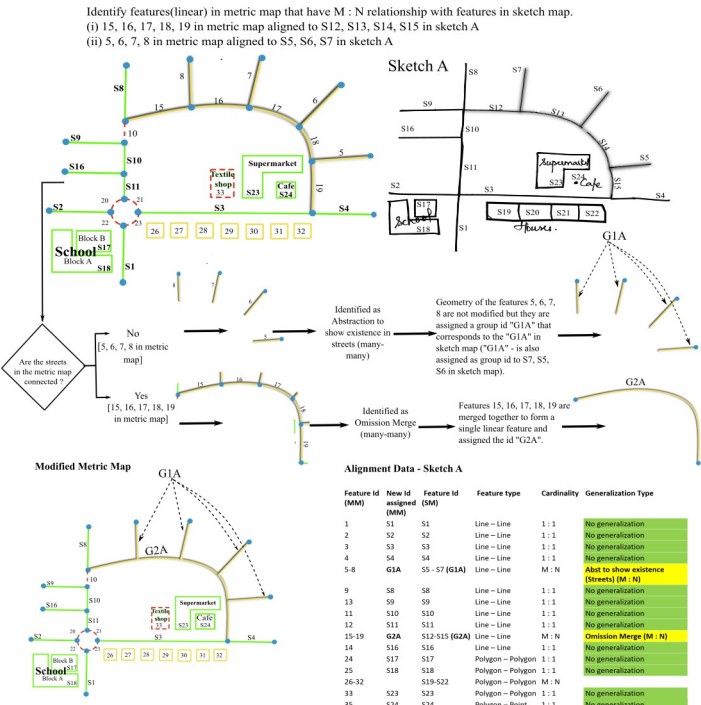

**Fig 6. Algorithm to resolve Omission-merge (many—many) and Abstraction to show existence in streets (many–many).**

If more than one **disconnected** street segment in the sketch map is aligned to more than one **disconnected** street segment in the metric map, it is identified as ***Abstraction to show existence in streets (many-many)***. The geometry of the street segments is not modified and they are treated as a group of features aligned to another group of features by assigning a group ID (Fig 6).

- **Special Cases:**
  While most of the generalizations could be detected using the cardinal relationship between aligned features and their feature type, more heuristics identifying Junction-merge and Roundabout-collapse are required.

  - **Roundabout-collapse:** Fig 7 shows how the street segments are aligned in the case of Roundabout-collapse. It can be seen that street segments S1, S2, S3 and S11 have a one-to-one match whereas street segments 20, 21, 22 and 23 are missing in the sketch map. Thus, the streets S1, S2, S3 and S11 would have been classified as no generalization initially based on the cardinal relationship between aligned features (1:1). To detect Roundabout-collapse, from the missing streets, streets with curvature are segregated. Once, we have the curved streets, the algorithm tries to detect if some of the curved streets form a closed loop or a polygon in metric map. If a closed loop or a polygon is present, then the streets touching the closed loop/polygon are segregated. If the corresponding street segments in the sketch map form a junction, the algorithm detects it as a case of ***Roundabout-collapse*** and the closed loop/polygon is replaced by its center point. The vertices of the street segments S1, S2, S3 and S11 are extended till this center point forming a junction instead of the roundabout.

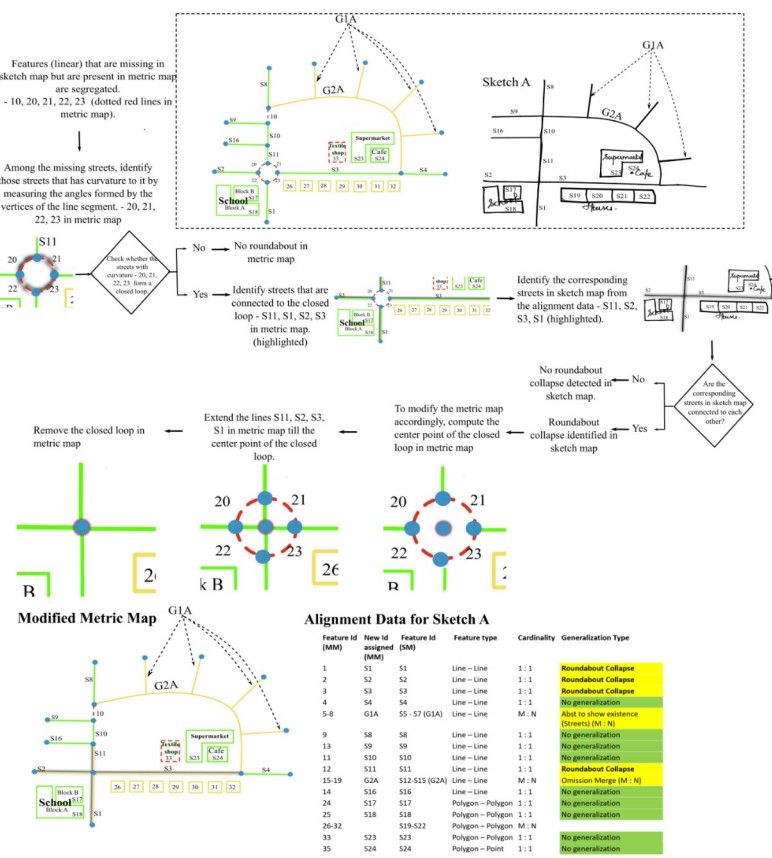

**Fig 7. Algorithm to resolve Roundabout-collapse.**

*3.2.1.1 Junction-merge.* Fig 8 shows how the street segments (S11,S2, S3, S1) are aligned in the case of a junction merge. It is a one-to-one alignment in linear features labeled as no generalization initially. To detect Junction-merge, one needs to compare the entire street network in a metric map with the entire street network in the sketch map. Having a closer look at the Junction-merge in Fig 8, it can be assumed that it is the missing street '10' in metric map, that causes the two junctions to merge. Hence, the streets that are present in the metric map but are missing in the sketch map are identified. Subsequently, we identify other streets in the metric map that are connected to this missing street. If the corresponding streets in the sketch map form a single junction, the algorithm identifies it as a *Junction-merge*. The two junctions are merged into one by replacing the missing street with its center point, followed by connecting the endpoints of other streets that formed two junctions previously to this center point, resulting in a single junction.

**3.2.2 Generalization in buildings.** In our previous work, we identified three types of generalization in buildings: Collapse, Amalgamation, and Abstraction to show existence.

- **Subcase: Many-to-one.** If one polygon in a sketch map is aligned with many polygons in the metric map, then the algorithm identifies it as a case of *Amalgamation*. If the generalization is classified as an Amalgamation, we compute a convex hull on the multiple polygons aligned in the metric map to arrive at a single polygon (Fig 9). This polygon then has a one-to-one relationship with the polygon in the sketch map for further analysis.

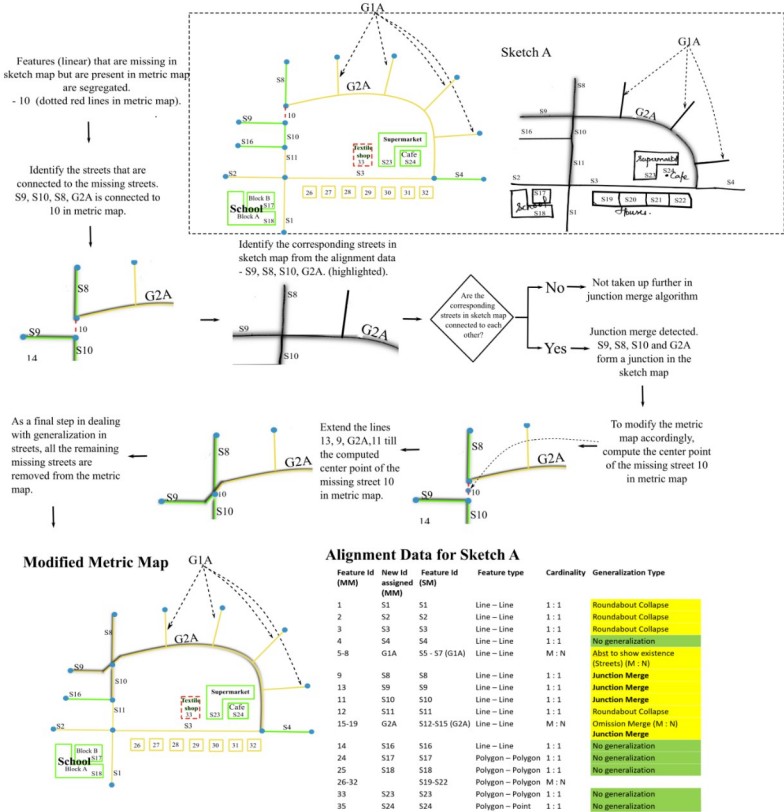

**Fig 8. Algorithm to resolve Junction-merge.**

- **Subcase: Many-to-many**. If many polygons in a sketch map are aligned with many polygons in the metric map, then the algorithm identifies it as **Abstraction to show existence in buildings**. No geometric modifications are made to the features in the metric map (Fig 10) but a group ID is assigned to the group of features in both metric map and sketch map.

- **Special Case:** A polygonal feature in a metric map aligned to a point feature in a sketch map is detected as **Collapse**. The center point of the polygon under consideration in the metric map is computed which replaces the polygon (Fig 11).

As a final output, the algorithm produces two different generalized versions of the original metric map, each corresponding to the two sketch maps–Sketch A and Sketch B. Note that the aligned features in the metric map and sketch map share the same id in Fig 12.

### 3.3 Implementation of the algorithm

We implemented our algorithm as an online tool in SketchMapia [12], an ongoing project focusing on the analysis of sketch maps. SketchMapia has provisions for digitizing and aligning metric maps and sketch maps. The features of interest in the metric map are decided by the researcher manually depending on their experimental setup used to collect sketch maps. Once features are digitized and aligned in SketchMapia, our online tool can be utilized to detect generalization in sketch maps. Our tool produces four different types of output:

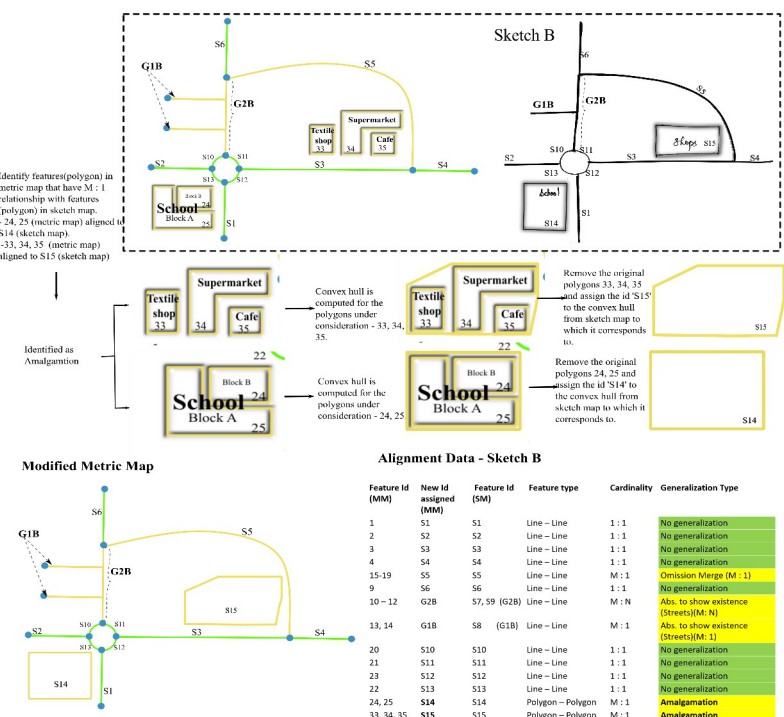

**Fig 9. Algorithm to resolve Amalgamation.**

- It produces a list containing the number of times a particular generalization type has been identified in a sketch map.

- It produces a downloadable CSV file containing the information on which feature IDs in the metric map are mapped to which feature IDs in the sketch map along with the identified generalization type that each feature has undergone.

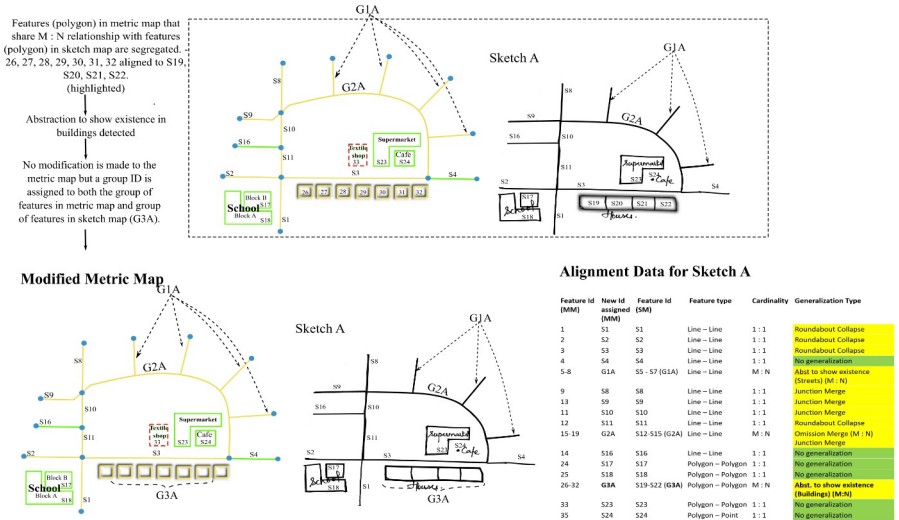

**Fig 10. Algorithm to resolve Abstraction to show existence in buildings.**

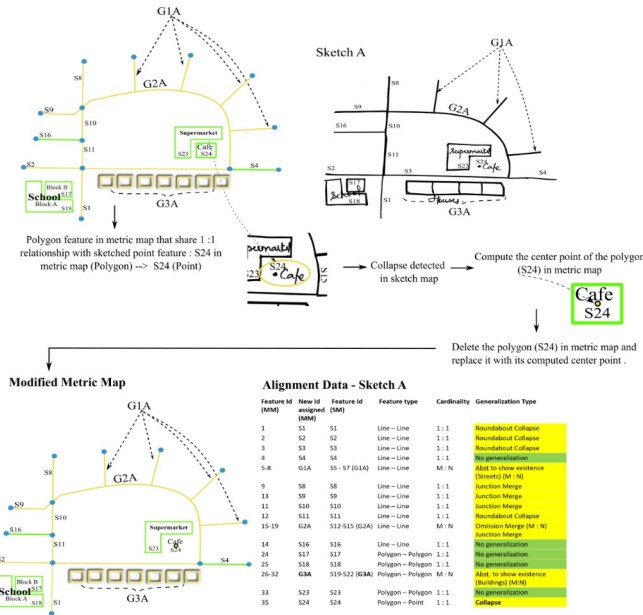

**Fig 11. Algorithm to resolve collapse.**

- It produces a color-coded sketch map and metric map that highlights the generalized features as yellow and non-generalized features as green in the sketch map; missing features as red in the metric map.

- It results in a geometrically modified generalized metric map that has a one-on-one correspondence with a sketch map for further analysis.

(a) Metric map generated for sketch A

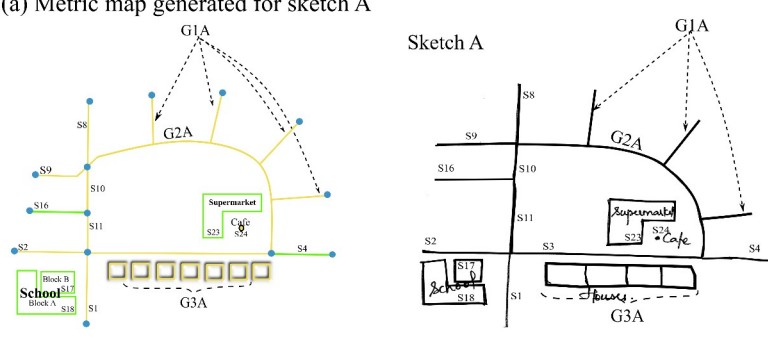

(b) Metric map generated for sketch B

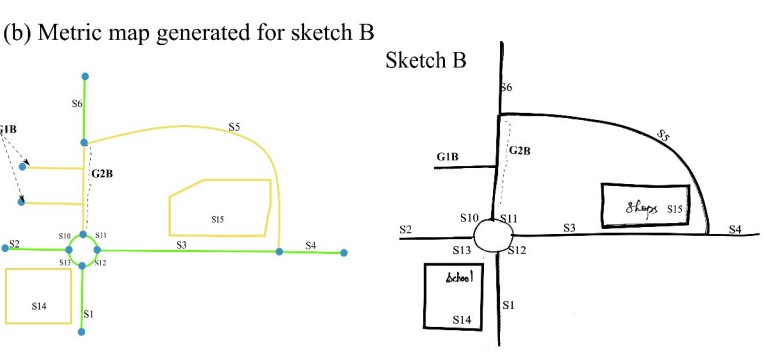

**Fig 12. Generalized metric map produced by the algorithm for sketch A (a) and sketch B (b).**

## 4 Evaluation of the algorithm

We evaluated our algorithm to check if it detects all generalizations in a sketch map by comparing the results from our online tool to that of the results from our previous study conducted using human raters [11]. We assume that having a high agreement with our human raters implies that the algorithm can detect the generalizations in sketch maps of various spatial configurations consistently.

### 4.1 Data

We used the same set of 11 sketch maps and their corresponding metric maps from our previously conducted user study [11]. The sketch maps corresponded to four different real-world locations. The 11 sketch maps had 328 features in total with 250 non-generalized and 78 generalized features. As our algorithm also detects omission merge (many-many), we include additional 6 generalizations in the present study resulting in a total of 84 generalized features.

### 4.2 Method

Sketch maps and their corresponding metric map were digitized to translate them into vector format. We aligned the features in the sketch map to those in the metric map using the functionalities available in SketchMapia. Our online tool resulted in four different outputs as mentioned in section 3.3. The same procedure was repeated for all ten sketch maps. We focused on two of the outputs: a color-coded sketch map and the CSV file containing information on which features in the sketch map have been aligned to which features in their metric map, as well as their generalization type. As the other two outputs from the web application are based on this CSV file, we assumed the reliability of the CSV file would also ensure the reliability of the other two outputs. The color-coded map from the web application along with the output CSV file was compared to that of the color coding done by the participants in our previous study wherein they classified sketched features into generalized (yellow), non-generalized (green) or missing (red). Fig 13 shows the color-coded maps from our online tool along with the maps marked by the five participants for one of the sketch maps. The following considerations were made while analyzing the differences between the algorithm and human raters:

- The generalization types: Omission Merge and Omission Merge (many-many) were not directly marked by the participants. These two generalization types are consequences of missing streets and generalized side streets respectively. Hence, if a participant has marked a feature as missing in the metric map, and if the missing street leads to the merging of streets, we perceive that omission merge has been identified by participants.

- Similarly, if participant has identified abstraction to show existence in side streets and if this leads to merging of the main streets, we perceive that omission merge (many-many) has been identified by the participants.

### 4.3 Results

Table 2 shows the generalization types introduced in each of the sketch maps along with whether they were identified by the five participants and the algorithm.

The numbers 0 to 10 in Table 2 represents the different sketch maps. Within each sketch map, a number of generalizations were introduced. For instance, sketch map 0 had two instances of amalgamation, one instance of collapse, one instance of abstraction to show existence in buildings, one instance of roundabout collapse and one instance of omission merge.

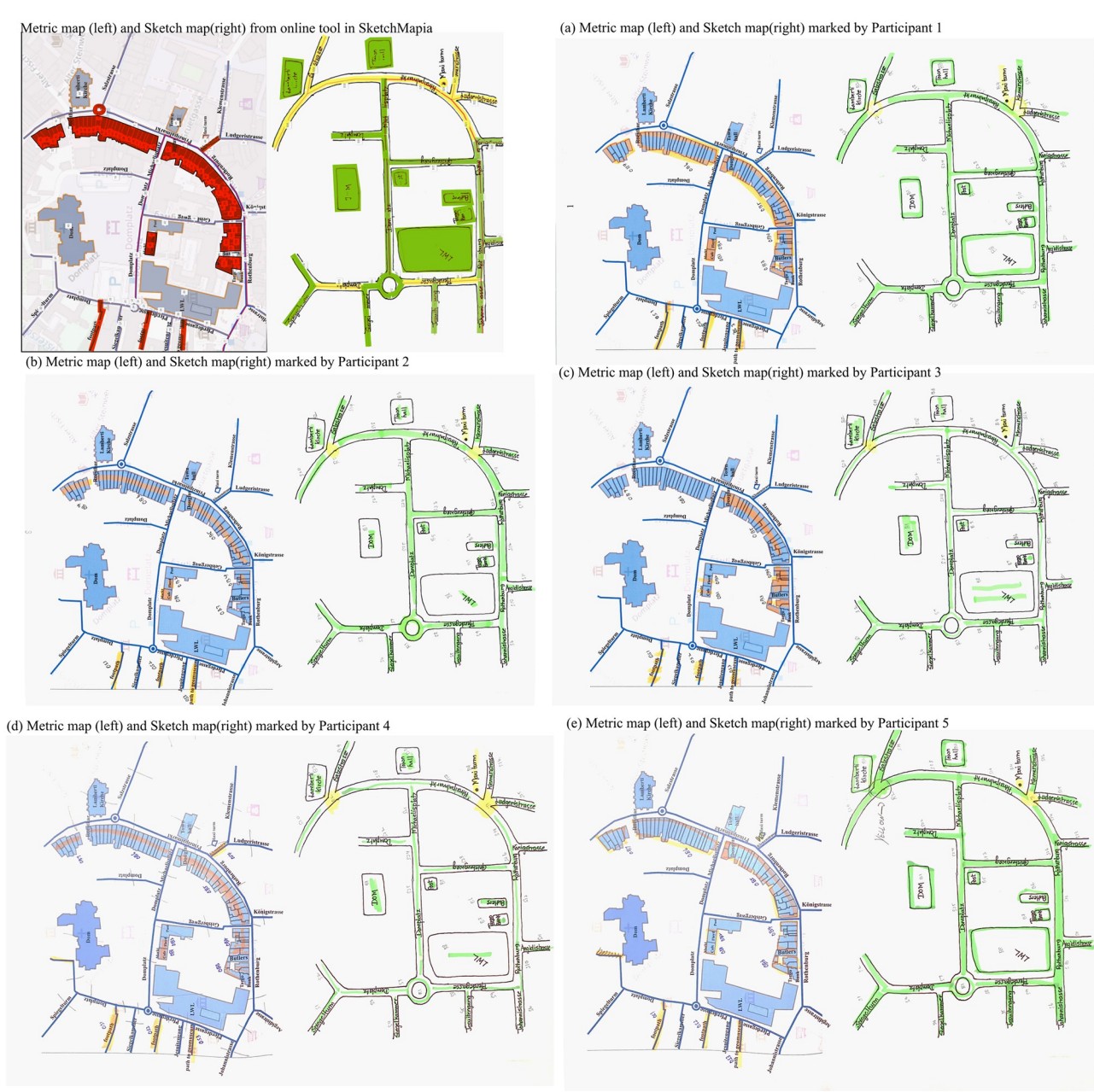

Metric map (left) and Sketch map(right) from online tool in SketchMapia

(a) Metric map (left) and Sketch map(right) marked by Participant 1

(b) Metric map (left) and Sketch map(right) marked by Participant 2

(c) Metric map (left) and Sketch map(right) marked by Participant 3

(d) Metric map (left) and Sketch map(right) marked by Participant 4

(e) Metric map (left) and Sketch map(right) marked by Participant 5

**Fig 13. Comparison of output from the online tool implementing the algorithm and human raters.**

We computed the percentage agreement between our web application and that of the majority of the participants in identifying generalization as shown in Table 3. Our algorithm has 100% agreement with the majority in nine out of the eleven sketch maps. In sketch maps 5 and 9, the agreement was 88.8% and 83.3% respectively as participants failed to identify generalization in these cases. From Table 2, it can be seen that in sketch map 5, omission merge has not been identified by the majority of participants. In sketch map 9, junction merge has not been identified by any of the participants. When examining the data on a participant-by-participant basis, it is evident that none of the participants were able to identify all of the 84 generalizations.

**Table 2. Generalizations in sketch maps(SM) identified by participants (P1,P2,P3,P4 and P5) and our online tool (OT).** Tick mark indicates that the particular generalization was identified and a cross mark indicates that the generalization was not identified.

| SM | Generalization | P1 | P2 | P3 | P4 | P5 | OT | SM | Generalization | P1 | P2 | P3 | P4 | P5 | OT |
|---|---|---|---|---|---|---|---|---|---|---|---|---|---|---|---|
| 0 | (i) Amalgamation | ✓ | ✓ | ✓ | ✓ | ✓ | ✓ | 4 | (viii) Junction Merge | ✓ | ✓ | × | ✓ | ✓ | ✓ |
| 0 | (ii) Amalgamation | ✓ | ✓ | ✓ | ✓ | ✓ | ✓ | 5 | (i) Abst to show existence in streets | ✓ | ✓ | × | ✓ | ✓ | ✓ |
| 0 | (iii) Collapse | ✓ | ✓ | ✓ | ✓ | ✓ | ✓ | 5 | (ii) Omission Merge (many-many) | ✓ | ✓ | × | ✓ | ✓ | ✓ |
| 0 | (iv) Abst. to show existence in buildings | ✓ | ✓ | ✓ | ✓ | ✓ | ✓ | 5 | (iii) Abst to show existence in buildings | ✓ | ✓ | ✓ | ✓ | ✓ | ✓ |
| 0 | (v) Roundabout Collapse | ✓ | ✓ | ✓ | ✓ | ✓ | ✓ | 5 | (iv) Roundabout Collapse | ✓ | ✓ | ✓ | ✓ | ✓ | ✓ |
| 0 | (vi) Omission Merge | ✓ | ✓ | ✓ | ✓ | ✓ | ✓ | 5 | (v) Junction Merge | ✓ | ✓ | ✓ | × | ✓ | ✓ |
| 1 | (i) Omission Merge | ✓ | ✓ | ✓ | ✓ | ✓ | ✓ | 5 | (vi) Omission Merge | ✓ | ✓ | ✓ | × | ✓ | ✓ |
| 1 | (ii) Omission Merge | ✓ | ✓ | ✓ | ✓ | ✓ | ✓ | 5 | (vii) Omission Merge | × | ✓ | ✓ | × | ✓ | ✓ |
| 1 | (iii) Omission Merge | ✓ | ✓ | ✓ | ✓ | ✓ | ✓ | 5 | (viii) Omission Merge | × | × | ✓ | × | × | ✓ |
| 1 | (iv) Junction Merge | ✓ | ✓ | × | ✓ | ✓ | ✓ | 6 | (i) Amalgamation | ✓ | ✓ | ✓ | ✓ | ✓ | ✓ |
| 1 | (v) Collapse | ✓ | ✓ | ✓ | ✓ | ✓ | ✓ | 6 | (ii) Collapse | ✓ | ✓ | ✓ | ✓ | ✓ | ✓ |
| 1 | (vi) Roundabout Collapse | ✓ | ✓ | ✓ | ✓ | ✓ | ✓ | 6 | (iii) Amalgamation | ✓ | ✓ | × | ✓ | ✓ | ✓ |
| 2 | (i) Junction Merge | ✓ | ✓ | ✓ | ✓ | × | ✓ | 6 | (iv) Junction Merge | ✓ | ✓ | ✓ | × | ✓ | ✓ |
| 2 | (ii) Omission Merge | ✓ | × | ✓ | ✓ | × | ✓ | 6 | (v) Junction Merge | ✓ | ✓ | ✓ | ✓ | × | ✓ |
| 2 | (iii) Abst. to show existence in streets | ✓ | ✓ | ✓ | ✓ | ✓ | ✓ | 6 | (vi) Omission Merge | ✓ | ✓ | ✓ | ✓ | ✓ | ✓ |
| 2 | (iv) Junction Merge | ✓ | ✓ | ✓ | ✓ | ✓ | ✓ | 6 | (vii) Omission Merge | ✓ | ✓ | ✓ | ✓ | ✓ | ✓ |
| 2 | (v) Junction Merge | ✓ | ✓ | × | ✓ | × | ✓ | 6 | (viii) Omission Merge | ✓ | ✓ | ✓ | ✓ | ✓ | ✓ |
| 2 | (vi) Amalgamation | × | ✓ | × | ✓ | ✓ | ✓ | 6 | (ix) Omission Merge | ✓ | ✓ | ✓ | ✓ | ✓ | ✓ |
| 2 | (vii) Abst. to show existence in buildings | ✓ | ✓ | ✓ | ✓ | ✓ | ✓ | 6 | (x) Omission Merge | ✓ | ✓ | ✓ | ✓ | ✓ | ✓ |
| 2 | (viii) Abst. to show existence in buildings | ✓ | ✓ | ✓ | ✓ | ✓ | ✓ | 7 | (i) Amalgamation | ✓ | ✓ | × | ✓ | ✓ | ✓ |
| 2 | (ix) Roundabout Collapse | ✓ | ✓ | ✓ | ✓ | ✓ | ✓ | 7 | (ii) Amalgamation | ✓ | ✓ | ✓ | ✓ | ✓ | ✓ |
| 2 | (x) Roundabout Collapse | ✓ | ✓ | ✓ | ✓ | ✓ | ✓ | 7 | (iii) Roundabout Collapse | ✓ | ✓ | ✓ | ✓ | ✓ | ✓ |
| 2 | (xi) Omission Merge (many-many) | ✓ | ✓ | ✓ | ✓ | ✓ | ✓ | 7 | (iv) Omission Merge | × | ✓ | ✓ | ✓ | ✓ | ✓ |
| 2 | (xii) Omission Merge (many-many) | ✓ | ✓ | ✓ | ✓ | ✓ | ✓ | 7 | (v) Omission Merge | ✓ | ✓ | ✓ | ✓ | ✓ | ✓ |
| 3 | (i) Omission Merge | ✓ | ✓ | ✓ | ✓ | ✓ | ✓ | 7 | (vi) Omission Merge | ✓ | ✓ | ✓ | ✓ | ✓ | ✓ |
| 3 | (ii) RoundAboutCollapse | ✓ | ✓ | ✓ | ✓ | ✓ | ✓ | 7 | (vii) Omission Merge | ✓ | ✓ | ✓ | ✓ | ✓ | ✓ |
| 3 | (iii) Omission Merge | ✓ | ✓ | ✓ | ✓ | ✓ | ✓ | 7 | (viii) Omission Merge | ✓ | ✓ | ✓ | ✓ | ✓ | ✓ |
| 3 | (iv) Omission Merge | ✓ | ✓ | ✓ | ✓ | ✓ | ✓ | 7 | (ix) Omission Merge | ✓ | ✓ | ✓ | ✓ | ✓ | ✓ |
| 3 | (v) Collapse | ✓ | ✓ | ✓ | ✓ | ✓ | ✓ | 8 | (i) Abst to show existence in streets | ✓ | ✓ | ✓ | ✓ | ✓ | ✓ |
| 3 | (vi) Collapse | ✓ | ✓ | ✓ | ✓ | ✓ | ✓ | 8 | (ii) Omission Merge (many-many) | ✓ | ✓ | ✓ | ✓ | ✓ | ✓ |
| 3 | (vii) Collapse | ✓ | ✓ | ✓ | ✓ | ✓ | ✓ | 8 | (iii) Amalgamation | ✓ | ✓ | × | ✓ | ✓ | ✓ |
| 3 | (viii) Collapse | ✓ | ✓ | ✓ | ✓ | ✓ | ✓ | 8 | (iv) Amalgamation | ✓ | ✓ | ✓ | ✓ | ✓ | ✓ |
| 3 | (ix) Omission Merge | ✓ | ✓ | ✓ | ✓ | ✓ | ✓ | 8 | (v) Abst to show existence in buildings | ✓ | ✓ | ✓ | ✓ | ✓ | ✓ |
| 3 | (x) Omission Merge | ✓ | ✓ | ✓ | ✓ | ✓ | ✓ | 8 | (vi) Abst to show existence in buildings | ✓ | ✓ | ✓ | ✓ | ✓ | ✓ |
| 3 | (xi) Omission Merge | ✓ | ✓ | × | × | ✓ | ✓ | 9 | (i) Abst to show existence in streets | ✓ | ✓ | ✓ | ✓ | ✓ | ✓ |
| 4 | (i) Abst. to show existence in streets | ✓ | ✓ | ✓ | ✓ | ✓ | ✓ | 9 | (ii) Omission Merge (many-many) | ✓ | ✓ | ✓ | ✓ | ✓ | ✓ |
| 4 | (ii) Omission Merge (many-many) | ✓ | ✓ | ✓ | ✓ | ✓ | ✓ | 9 | (iii) Amalgamation | ✓ | ✓ | × | ✓ | ✓ | ✓ |
| 4 | (iiii) Abst. to show existence in buildings | ✓ | ✓ | ✓ | ✓ | ✓ | ✓ | 9 | (iv) Amalgamation | ✓ | ✓ | ✓ | ✓ | ✓ | ✓ |
| 4 | (iv) Abst to show existence in buildings | ✓ | ✓ | ✓ | ✓ | ✓ | ✓ | 9 | (v) Roundabout Collapse | ✓ | ✓ | ✓ | ✓ | ✓ | ✓ |
| 4 | (v) Abst to show existence in buildings | ✓ | ✓ | ✓ | ✓ | ✓ | ✓ | 9 | (vi) Junction Merge | × | × | × | × | × | ✓ |
| 4 | (vi) Roundabout Collapse | ✓ | ✓ | ✓ | ✓ | ✓ | ✓ | 10 | (i) Amalgamation | ✓ | ✓ | ✓ | ✓ | ✓ | ✓ |
| 4 | (vii) Roundabout Collapse | ✓ | ✓ | ✓ | ✓ | ✓ | ✓ | 10 | (ii) Collapse | ✓ | ✓ | ✓ | ✓ | ✓ | ✓ |

**Table 3. Percentage agreement of the results from our online tool to the majority of the participants.**

| Sketch Map | No. of generalization | Cases identified by majority of participants | Cases identified by our online tool | Percentage agreement |
|---|---|---|---|---|
| SM 0 | 6 | 6 | 6 | 100% |
| SM 1 | 6 | 6 | 6 | 100% |
| SM 2 | 12 | 12 | 12 | 100% |
| SM 3 | 11 | 11 | 11 | 100% |
| SM 4 | 8 | 8 | 8 | 100% |
| SM 5 | 8 | 7 | 8 | 87.5% |
| SM 6 | 10 | 10 | 10 | 100% |
| SM 7 | 9 | 9 | 9 | 100% |
| SM 8 | 6 | 6 | 6 | 100% |
| SM 9 | 6 | 5 | 6 | 83.3% |
| SM 10 | 2 | 2 | 2 | 100% |

We also looked at the total number of generalization instances introduced in the sketch map and the percentage identified by participants and our algorithm. Table 4 shows that while some generalization types such as junction merge was identified the least number of times-71% of the total instances introduced, our algorithm identified every instance of it. In the case of other generalizations such as Roundabout Collapse, Abstraction to show existence in buildings and Collapse, all of the introduced instances were identified by both the participants and our online tool.

## 4.4 Discussion

The following observations can be made from the evaluation of our algorithm.

The differences in sketch map 5 and 9 are not due to an error in the implementation but rather due to participants failing to consistently color code the sketch maps. Sketch maps easily become too complex for people, and thus participants sometimes forgot to mark all the features in the sketch map as either green or yellow sometimes leaving them unclassified. These errors happen in a manual process.

It was observed that not all generalizations were identified equally by the participants: junction merge was identified the lowest number of times (Table 4). In sketch map 9, none of the participants identified a junction merge, while our algorithm detected it (Table 2). This indicates that our algorithm is consistent in its strategy to detect generalizations even in

**Table 4. Percentage of generalization instances identified by the participants and the online tool.**

| Generalization Type | Total no. of instances in 11 sketch maps | Percentage identified by participants | Percentage identified by our online tool |
|---|---|---|---|
| Omission that leads to merging of streets | 25 | 95.2% | 100% |
| Abstraction to show existence in streets | 5 | 96% | 100% |
| Junction Merge | 9 | 71.1% | 100% |
| Roundabout Collapse | 10 | 100% | 100% |
| Amalgamation | 12 | 88% | 100% |
| Abstraction to show existence in buildings | 9 | 100% | 100% |
| Collapse | 8 | 100% | 100% |
| Omission Merge (many-many) | 6 | 96.6% | 100% |
| Total | 84 | 93.3% | 100% |

complicated networks of streets, unlike human raters who either fail to notice such complex generalizations or change their strategy when moving from one sketch map to the other.

Some of the generalizations such as Abstraction to show existence in streets and Omission-merge have more than one consequence in the resulting map. While participants are able to identify the generalization, they fail to observe its consequences. Our algorithm can not only detect a generalization but also capture all its consequences. This is also one of the reasons we see a difference between the implementation and human raters in sketch map 5.

Regarding the computational efficiency of our algorithm, categorization of all the sketched features based on its feature type and further sub-categorization of it based on cardinality of the aligned features enables parallel processing to resolve generalization in sketch map. This parallelization significantly reduces the time complexity as the algorithm need not wait till it resolves the generalization of one feature before moving on to the next. In terms of space complexity, both the input–metric map, and sketch map and the resulting generalized map are in GeoJSON format as required by the online tool- SketchMapia. As GeoJSON format is not the most efficient way of storing spatial data due to the lack of indexing of features and has larger file size compared to the other spatial formats such as shapefile, there is still room for increasing the efficiency if very large sketch maps need to be analyzed. The algorithm can be implemented to accept input in shapefile format, to expedite the process. On the other hand, sketch maps are not likely to have more than 30 to 50 features, minimizing the performance concerns.

## 5 Conclusion

Comparing sketch maps to metric maps and analyzing their quality has been a challenge, in particular because features in sketch maps turned out to be generalized and thus not easily comparable to corresponding features in the metric map. This paper presents a new algorithmic approach to compare generalized sketch maps to a metric map by (i) automatically detecting the type of generalization, (ii) transforming the features such that corresponding features in maps can be analyzed.

We compared the software results to expert ratings. The high percentage of agreement in identifying the generalization types with our human raters shows the reliability of our algorithm in detecting generalization in sketch maps. In addition to detecting generalization in the sketch map, it also generates a metric map with most of the features having 1:1 correspondence with features in the sketch map. This would further reduce the information loss occurring due to a lack of procedure to deal with generalized content. Though the evaluation of the algorithm is done using systematically created sketch maps, our algorithm can detect generalization in sketch maps from real-world experiments as well.

We integrated our online tool with SketchMapia, a sketch map analysis software. Previously, analyzing sketch maps using SketchMapia required manually digitizing the metric map for each sketch map, a time-consuming process. Researches often had to repeat this process for tens or hundreds of sketch maps, leading to errors and inefficiency. Our tool streamlines this task by digitizing the metric map only once, regardless of the number of sketch maps. Our algorithm generates different metric maps corresponding to each sketch map from the original map and alignment data. This approach can also be applied to other software like Gardony map drawing analyzer, addressing similar challenges with generalization handling. There are also use cases other than sketch map analysis which would benefit from our algorithm. In studies related to qualitative constraint networks for sketch maps, the proposed algorithm would enable qualitative matching between metric maps and sketch maps at different levels of generalization. When combined with participatory GIS methods, our algorithm can help to

integrate data from different sketch maps having different levels of generalization in a consistent manner. We believe that our algorithm to resolve generalization in sketch maps would be a valuable addition to a wide range of applications that utilize sketch maps.

## Author Contributions

**Conceptualization:** Charu Manivannan, Jakub Krukar, Angela Schwering.

**Data curation:** Charu Manivannan.

**Formal analysis:** Charu Manivannan.

**Funding acquisition:** Angela Schwering.

**Investigation:** Charu Manivannan.

**Methodology:** Jakub Krukar, Angela Schwering.

**Project administration:** Angela Schwering.

**Software:** Charu Manivannan.

**Supervision:** Jakub Krukar, Angela Schwering.

**Validation:** Charu Manivannan.

**Visualization:** Charu Manivannan.

**Writing – original draft:** Charu Manivannan.

**Writing – review & editing:** Jakub Krukar, Angela Schwering.

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
