## [Decision Letter · Decision Letter 0]

31 Jan 2024

PONE-D-23-41484An Algorithmic Approach to Detect Generalization in Sketch Maps from Sketch Map AlignmentPLOS ONE

Dear Dr. Manivannan,

Thank you for submitting your manuscript to PLOS ONE. After careful consideration, we feel that it has merit but does not fully meet PLOS ONE’s publication criteria as it currently stands. Therefore, we invite you to submit a revised version of the manuscript that addresses the points raised during the review process.

Reviewer #1: The research presented holds substantial scientific merit, particularly in the domains of Geoscience and Digital Charts, and offers innovative insights. Its originality is commendable, and the availability of the code through a GitHub repository is appreciated.

However, there are areas that require attention:

The manuscript's length is excessive, leading to reader confusion. A more concise presentation, especially regarding generalization approaches, would enhance readability. These approaches, already discussed in prior studies, could be summarized briefly.

There is noticeable repetition within the text. For example, the content description of Figure 1 is redundant on pages 6 and 8.

Specific concerns and suggestions include:

1) The detailed description of generalization techniques (e.g., on page 20) is overly extensive. A brief summary with references to existing literature would suffice.

2) The method for identifying regions of interest from sketch drawings needs clarification. It's unclear whether this requires manual input or is automatically detected. In other words, how do you specify OSM map region based from sketch map?

3) The computational complexity of the proposed method is not addressed. An analytical discussion of its feasibility for large datasets (1000+ features) would be beneficial.

4) My biggest concern is that the Table 2's content and the abbreviation 'SM' are unclear. Furthermore, there are repetitive row names, such as 'SMO 0 - Amalgamation,' which hinder the interpretation of results.

5) Chapter 5 appears to add little value. Consider removing or integrating it with Section 6 for brevity and clarity.

Overall, the study is a valuable contribution and seems fit for publication upon revision. I recommend focusing on Section 4, elaborating on the results and providing a more detailed discussion. A minor note: mentioning raster generalization may be unnecessary, as sketch maps inherently involve vector data.

I look forward to reviewing a revised version of the manuscript.

Reviewer #2: An Algorithmic Approach to Detect Generalization in Sketch Maps from Sketch Map Alignment

This is an interesting topic. The idea and contributions of this study appear to be a valid one, and together with their earlier work, it has the potential to offer a helpful guide for various disciplines. On the whole, the paper is structured reasonably, and its intentions and argument are relatively clear.

The discussion and conclusion sections are potentially the most interesting parts of the paper, and at present the least well-developed. A helpful way forward would be to identify a hierarchy to structure the implication more effectively.

It would be better for a new iteration of this paper to highlight the similarities and differences of this research compared to previous studies such as the following, and to specifically locate new concepts in the form of a review analysis.

Finally, major revision is suggested.

- Hadlos, A., Opdyke, A., & Hadigheh, S. A. (2022). Where does local and indigenous knowledge in disaster risk reduction go from here? A systematic literature review. International Journal of Disaster Risk Reduction, 103160.

- Membele, G. M., Naidu, M., & Mutanga, O. (2022). Examining flood vulnerability mapping approaches in developing countries: A scoping review. International Journal of Disaster Risk Reduction, 69, 102766.

- Klonner, C., Hartmann, M., Dischl, R., Djami, L., Anderson, L., Raifer, M., ... & Porto de Albuquerque, J. (2021). The sketch map tool facilitates the assessment of OpenStreetMap data for participatory mapping. ISPRS International Journal of Geo-Information, 10(3), 130.

- Shen, Y., Ai, T., Li, J., Huang, L., & Li, W. (2020). A progressive method for the collapse of river representation considering geographical characteristics. International Journal of Digital Earth, 13(12), 1366-1390.

- Jakobi, Á., & Pődör, A. (2020). GIS-based statistical analysis of detecting fear of crime with digital sketch maps: A Hungarian multicity study. ISPRS International Journal of Geo-Information, 9(4), 229.

- Brandt, K., Graham, L., Hawthorne, T., Jeanty, J., Burkholder, B., Munisteri, C., & Visaggi, C. (2020). Integrating sketch mapping and hot spot analysis to enhance capacity for community‐level flood and disaster risk management. The Geographical Journal, 186(2), 198-212.

We look forward to receiving your revised manuscript.

Kind regards,

Saeid Norouzian-Maleki, Ph.D.

Academic Editor

PLOS ONE

Journal Requirements:

"This work was supported by the German Research Foundation (SCHW1372/7-3, “Sketchmapia”) and the Swiss National Science Foundation (Sinergia 202284, “3D Sketch Maps”)."

4. We noted in your submission details that a portion of your manuscript may have been presented or published elsewhere. [Previously, we conducted a user study wherein human raters identified generalizations in sketch map. The results of which are published in Journal of environmental Psychology. In this article, we present an algorithmic approach that automatically detects generalization from sketch map alignment. We compare the output from the algorithm to the results of our previous study. As the main contribution of the paper is the algorithm, and we are using the data from previous study only for comparison, we request not to consider this as dual publication.] Please clarify whether this [conference proceeding or publication] was peer-reviewed and formally published. If this work was previously peer-reviewed and published, in the cover letter please provide the reason that this work does not constitute dual publication and should be included in the current manuscript.

Reviewers' comments:

Reviewer's Responses to Questions

**Comments to the Author**

1. Is the manuscript technically sound, and do the data support the conclusions?

Reviewer #1: Yes

Reviewer #2: Partly

2. Has the statistical analysis been performed appropriately and rigorously? 

Reviewer #1: No

Reviewer #2: Yes

3. Have the authors made all data underlying the findings in their manuscript fully available?

Reviewer #1: Yes

Reviewer #2: Yes

4. Is the manuscript presented in an intelligible fashion and written in standard English?

Reviewer #1: Yes

Reviewer #2: Yes

5. Review Comments to the Author

Reviewer #1: The research presented holds substantial scientific merit, particularly in the domains of Geoscience and Digital Charts, and offers innovative insights. Its originality is commendable, and the availability of the code through a GitHub repository is appreciated.

However, there are areas that require attention:

The manuscript's length is excessive, leading to reader confusion. A more concise presentation, especially regarding generalization approaches, would enhance readability. These approaches, already discussed in prior studies, could be summarized briefly.

There is noticeable repetition within the text. For example, the content description of Figure 1 is redundant on pages 6 and 8.

Specific concerns and suggestions include:

1) The detailed description of generalization techniques (e.g., on page 20) is overly extensive. A brief summary with references to existing literature would suffice.

2) The method for identifying regions of interest from sketch drawings needs clarification. It's unclear whether this requires manual input or is automatically detected. In other words, how do you specify OSM map region based from sketch map?

3) The computational complexity of the proposed method is not addressed. An analytical discussion of its feasibility for large datasets (1000+ features) would be beneficial.

4) My biggest concern is that the Table 2's content and the abbreviation 'SM' are unclear. Furthermore, there are repetitive row names, such as 'SMO 0 - Amalgamation,' which hinder the interpretation of results.

5) Chapter 5 appears to add little value. Consider removing or integrating it with Section 6 for brevity and clarity.

Overall, the study is a valuable contribution and seems fit for publication upon revision. I recommend focusing on Section 4, elaborating on the results and providing a more detailed discussion. A minor note: mentioning raster generalization may be unnecessary, as sketch maps inherently involve vector data.

I look forward to reviewing a revised version of the manuscript.

Note to Editor and Authors:

Please be aware that English is not my mother tongue. As a result, in this review, I have generally refrained from assessing the language quality and style of the manuscript. My focus was primarily on the content and scientific aspects of the study.

Reviewer #2: An Algorithmic Approach to Detect Generalization in Sketch Maps from Sketch Map Alignment

Dear Editor,

Thank you very much for the opportunity to look at this paper with the above title. This is an interesting topic. The idea and contributions of this study appear to be a valid one, and together with their earlier work, it has the potential to offer a helpful guide for various disciplines. On the whole, the paper is structured reasonably, and its intentions and argument are relatively clear.

The discussion and conclusion sections are potentially the most interesting parts of the paper, and at present the least well-developed. A helpful way forward would be to identify a hierarchy to structure the implication more effectively.

It would be better for a new iteration of this paper to highlight the similarities and differences of this research compared to previous studies such as the following, and to specifically locate new concepts in the form of a review analysis.

Finally, major revision is suggested.

- Hadlos, A., Opdyke, A., & Hadigheh, S. A. (2022). Where does local and indigenous knowledge in disaster risk reduction go from here? A systematic literature review. International Journal of Disaster Risk Reduction, 103160.

- Membele, G. M., Naidu, M., & Mutanga, O. (2022). Examining flood vulnerability mapping approaches in developing countries: A scoping review. International Journal of Disaster Risk Reduction, 69, 102766.

- Klonner, C., Hartmann, M., Dischl, R., Djami, L., Anderson, L., Raifer, M., ... & Porto de Albuquerque, J. (2021). The sketch map tool facilitates the assessment of OpenStreetMap data for participatory mapping. ISPRS International Journal of Geo-Information, 10(3), 130.

- Shen, Y., Ai, T., Li, J., Huang, L., & Li, W. (2020). A progressive method for the collapse of river representation considering geographical characteristics. International Journal of Digital Earth, 13(12), 1366-1390.

- Jakobi, Á., & Pődör, A. (2020). GIS-based statistical analysis of detecting fear of crime with digital sketch maps: A Hungarian multicity study. ISPRS International Journal of Geo-Information, 9(4), 229.

- Brandt, K., Graham, L., Hawthorne, T., Jeanty, J., Burkholder, B., Munisteri, C., & Visaggi, C. (2020). Integrating sketch mapping and hot spot analysis to enhance capacity for community‐level flood and disaster risk management. The Geographical Journal, 186(2), 198-212.

6. PLOS authors have the option to publish the peer review history of their article (what does this mean?). If published, this will include your full peer review and any attached files.

Reviewer #1: **Yes: **Andrzej Chybicki, PhD Eng.

Reviewer #2: No

---

## [Author Response · Author response to Decision Letter 0]

3 Apr 2024

Dear Reviewers, 

 Thank you for your valuable comments. It has helped us to improve the quality of the paper significantly. Below you can find the reply addressed to each of the suggestions. 

Reviewer #1: Andrzej Chybicki, PhD Eng.

The manuscript's length is excessive, leading to reader confusion. A more concise presentation, especially regarding generalization approaches, would enhance readability. These approaches, already discussed in prior studies, could be summarized briefly.

The length of the manuscript has been reduced by four pages.

There is noticeable repetition within the text. For example, the content description of Figure 1 is redundant on pages 6 and 8.

The explanation of Figure 1 on page 8 has been removed.

Specific concerns and suggestions include:

1) The detailed description of generalization techniques (e.g., on page 20) is overly extensive. A brief summary with references to existing literature would suffice.

The description of generalization technique in section 2.2 has been moved to fit within Figure 2. The description on page 20 is however not the explanation of generalization type, but rather how the algorithm detects the different generalization types. We apologize for not being clear with it. The texts on page 20 have been slightly modified to make it clear that the texts are description of generalization algorithm and not of generalization operator.

2) The method for identifying regions of interest from sketch drawings needs clarification. It's unclear whether this requires manual input or is automatically detected. In other words, how do you specify OSM map region based from sketch map?

It requires manual input. We have mentioned it now in section 3.3.

3) The computational complexity of the proposed method is not addressed. An analytical discussion of its feasibility for large datasets (1000+ features) would be beneficial.

 We have added the aspect of computational comlexity in terms of time and storage in section 4.2

4) My biggest concern is that the Table 2's content and the abbreviation 'SM' are unclear. 

Furthermore, there are repetitive row names, such as 'SMO 0 - Amalgamation,' which hinder the interpretation of results.

We have modified the Table 2’s content to make it clear and also added explanation on Table 2’s content in section 4.1.

5) Chapter 5 appears to add little value. Consider removing or integrating it with Section 6 for brevity and clarity.

We have removed chapter 5 and integrated it with the conclusion in section 5 (previously section 6)

Overall, the study is a valuable contribution and seems fit for publication upon revision. I recommend focusing on Section 4, elaborating on the results and providing a more detailed discussion. A minor note: mentioning raster generalization may be unnecessary, as sketch maps inherently involve vector data.

We have removed the texts on raster generalization (section 2.3)

Reviewer #2: An Algorithmic Approach to Detect Generalization in Sketch Maps from Sketch Map Alignment

This is an interesting topic. The idea and contributions of this study appear to be a valid one, and together with their earlier work, it has the potential to offer a helpful guide for various disciplines. On the whole, the paper is structured reasonably, and its intentions and argument are relatively clear.

The discussion and conclusion sections are potentially the most interesting parts of the paper, and at present the least well-developed. A helpful way forward would be to identify a hierarchy to structure the implication more effectively.

Thanks for pointing this out. We have elaborated the discussion sections and conclusion sections. The implication has been made clear in the conclusion.

It would be better for a new iteration of this paper to highlight the similarities and differences of this research compared to previous studies such as the following, and to specifically locate new concepts in the form of a review analysis.

Finally, major revision is suggested.

- Hadlos, A., Opdyke, A., & Hadigheh, S. A. (2022). Where does local and indigenous knowledge in disaster risk reduction go from here? A systematic literature review. International Journal of Disaster Risk Reduction, 103160.

- Membele, G. M., Naidu, M., & Mutanga, O. (2022). Examining flood vulnerability mapping approaches in developing countries: A scoping review. International Journal of Disaster Risk Reduction, 69, 102766.

- Klonner, C., Hartmann, M., Dischl, R., Djami, L., Anderson, L., Raifer, M., ... & Porto de Albuquerque, J. (2021). The sketch map tool facilitates the assessment of OpenStreetMap data for participatory mapping. ISPRS International Journal of Geo-Information, 10(3), 130.

- Shen, Y., Ai, T., Li, J., Huang, L., & Li, W. (2020). A progressive method for the collapse of river representation considering geographical characteristics. International Journal of Digital Earth, 13(12), 1366-1390.

- Jakobi, Á., & Pődör, A. (2020). GIS-based statistical analysis of detecting fear of crime with digital sketch maps: A Hungarian multicity study. ISPRS International Journal of Geo-Information, 9(4), 229.

- Brandt, K., Graham, L., Hawthorne, T., Jeanty, J., Burkholder, B., Munisteri, C., & Visaggi, C. (2020). Integrating sketch mapping and hot spot analysis to enhance capacity for community‐level flood and disaster risk management. The Geographical Journal, 186(2), 198-212.

Thank you for the references. We have discussed how our approach of handling generalization in sketch map can enhance the previous works in section 2.1. We have integrated all your suggested references into the text.

---

## [Decision Letter · Decision Letter 1]

17 May 2024

An algorithmic approach to detect generalization in sketch maps from sketch map alignment

PONE-D-23-41484R1

Dear Dr. Manivannan,

We’re pleased to inform you that your manuscript has been judged scientifically suitable for publication and will be formally accepted for publication once it meets all outstanding technical requirements.

Kind regards,

Saeid Norouzian-Maleki, Ph.D.

Academic Editor

PLOS ONE

Additional Editor Comments (optional):

Reviewers' comments:

Reviewer's Responses to Questions

**Comments to the Author**

1. If the authors have adequately addressed your comments raised in a previous round of review and you feel that this manuscript is now acceptable for publication, you may indicate that here to bypass the “Comments to the Author” section, enter your conflict of interest statement in the “Confidential to Editor” section, and submit your "Accept" recommendation.

Reviewer #1: All comments have been addressed

Reviewer #2: All comments have been addressed

2. Is the manuscript technically sound, and do the data support the conclusions?

Reviewer #1: Yes

Reviewer #2: Yes

3. Has the statistical analysis been performed appropriately and rigorously? 

Reviewer #1: Yes

Reviewer #2: Yes

4. Have the authors made all data underlying the findings in their manuscript fully available?

Reviewer #1: Yes

Reviewer #2: Yes

5. Is the manuscript presented in an intelligible fashion and written in standard English?

Reviewer #1: Yes

Reviewer #2: Yes

6. Review Comments to the Author

Reviewer #1:

My suggestions to the manuscript have been addressed. I recommend this research to be published in PLOS ONE Journal.

I want to congratulate Authors for good and valuable work that provides interesting scientific input for scientific community.

PS Note to editor: as I'm not a native speaker I recommend the language of the manuscript to be verified by an expert.

Reviewer #2: (No Response)

7. PLOS authors have the option to publish the peer review history of their article (what does this mean?). If published, this will include your full peer review and any attached files.

Reviewer #1: **Yes: **Andrzej Chybicki

Reviewer #2: No

---

## [Editor Report · Acceptance letter]

22 May 2024

PONE-D-23-41484R1 

PLOS ONE

Dear Dr. Manivannan, 

I'm pleased to inform you that your manuscript has been deemed suitable for publication in PLOS ONE. Congratulations! Your manuscript is now being handed over to our production team.

Kind regards, 

on behalf of

Dr. Saeid Norouzian-Maleki 

Academic Editor

PLOS ONE